# Establishing Hedgehog Gradients during Neural Development

**DOI:** 10.3390/cells12020225

**Published:** 2023-01-05

**Authors:** Sara Douceau, Tanya Deutsch Guerrero, Julien Ferent

**Affiliations:** 1INSERM UMR-S 1270, F-75005 Paris, France; 2Institut du Fer à Moulin, INSERM, Sorbonne Univeristy, F-75005 Paris, France

**Keywords:** Sonic Hedgehog, morphogen, gradient, neurodevelopment

## Abstract

A morphogen is a signaling molecule that induces specific cellular responses depending on its local concentration. The concept of morphogenic gradients has been a central paradigm of developmental biology for decades. Sonic Hedgehog (Shh) is one of the most important morphogens that displays pleiotropic functions during embryonic development, ranging from neuronal patterning to axon guidance. It is commonly accepted that Shh is distributed in a gradient in several tissues from different origins during development; however, how these gradients are formed and maintained at the cellular and molecular levels is still the center of a great deal of research. In this review, we first explored all of the different sources of Shh during the development of the nervous system. Then, we detailed how these sources can distribute Shh in the surrounding tissues via a variety of mechanisms. Finally, we addressed how disrupting Shh distribution and gradients can induce severe neurodevelopmental disorders and cancers. Although the concept of gradient has been central in the field of neurodevelopment since the fifties, we also describe how contemporary leading-edge techniques, such as organoids, can revisit this classical model.

## 1. Introduction

During embryonic development, cells need to determine their precise location to acquire specific identities and organize the architecture of the tissue. Morphogens induce polarization and regionalization in many organs. Today, the definition of a morphogen is a secreted protein that acts away from its source in a concentration dependent manner. In the 1950s, Alan Turing was the first to develop the idea of a signal going from one population of cells to another in order to create different territories within the same initial group of cells [1]. A few years later, the emergence of cellular heterogeneity during development was still a great matter of debate. Lewis Wolpert introduced the famous “French flag problem”, wondering what mechanisms are at play to create these beautiful and defined patterns of different cell identities during development [2,3]. The first concept associated with this problem is obviously the action of morphogens, acting in a gradient of concentration and allowing cells to be exposed to different thresholds and, therefore, inducing the distinct identities. Since then, many experimental studies have demonstrated how different concentrations of a morphogen can differentially impact the outcome of the cellular response. However, this concentration gradient has to be created and maintained in a complex cell organization with cellular movements, tissue constraints, dynamics, and sometimes fluid flows. In addition to this, molecular parameters should also be considered, such as protein expression, transport, adhesion/interaction, recycling, and degradation [4].

Here, we tackle the question of morphogen distribution by focusing on Sonic Hedgehog (Shh). This morphogen has been studied for more than 30 years in various contexts from invertebrates to vertebrates during development and adulthood [5]. Shh is part of the Hedgehog family, which is composed of three genes in vertebrates: Sonic Hedgehog, Indian Hedgehog, and Desert Hedgehog. This gene family was first discovered in Drosophila via a genetic screen [6]. During vertebrate development, Shh is famous for its role in the ventral patterning of neurons in the neural tube [7]. It also has many other roles, such as axon guidance, cerebellum development, and adult neurogenesis [8]. Its ability to trigger such different cellular responses makes it an intriguing pleiotropic factor. Many laboratories focus their research on the different signaling pathways downstream of this morphogen; however, before inducing specific responses in the targeted cells, the protein needs to be distributed and addressed at precise locations and specific times. In this review, we concentrate on its secretion, diffusion, and roles in the central nervous system (CNS). By describing the results obtained from historical studies and highlighting recent studies using state-of-the-art techniques, such as organoids, -omics approaches, live-imaging, and mathematical modelling, we will show the major advances in the understanding of the Shh gradient set-up and the challenges that remain in this field.

Shh, like its paralogs, goes through several post-translational modifications, which are particularly relevant for its diffusion. Indeed, Shh is one of the rare proteins that is covalently attached to cholesterol [9]. This modification occurs as the precursor protein is cleaved in the endoplasmic reticulum. The cholesterol is added at the C-terminus of the N-terminal fragment of the precursor. Later on, a palmitoyl residue is added at the N-terminus of the N-terminal fragment, completing the final structure of the signal peptide [10]. Although these modifications are not required for the potency of the protein, they are essential for its diffusion at long ranges (as described more in detail in Section 3.1). The secretion of Shh is mediated via Dispatched, a large membrane protein which has been recently crystallized [11,12]. Ultimately, Shh binds to Patched1 (Ptch1; [13,14]) and to a coreceptor, which can either be Cdon, Brother of Cdon (Boc), or Gas1 (Figure 1; [15,16]. The dynamics of binding between Ptch1 and Shh regulates the formation of the morphogenic gradient itself [17]. In vertebrates, when Shh binds to Ptch1, it triggers its removal from a specific membrane structure called the primary cilium. The primary cilium is a unique membrane extension based on an axoneme composed of microtubules. The removal of Ptch1 from the primary cilia is concomitant with the targeting of the protein Smoothened (Smo) at the cilium (Figure 1A,B; [18]). There, Smo is activated and will allow the activation of the Gli transcription factor family at the tip of the cilia and eventually at the nucleus [19]. This will trigger the transcription of several target genes such as *Ptch1* itself. Overall, this signaling pathway is considered to be the canonical pathway in vertebrates (Figure 1C). Several other signaling pathways exist, which do not go through transcriptional responses or even Smo activation [8]. For instance, we can cite Ptch1’s role as a dependent receptor to induce apoptosis [20] or Shh’s non-canonical regulation of calcium activity [21], cell migration of fibroblasts [22], and axon guidance in the neural tube or visual system (as described more in detail in Section 2.1). Overall, this shows how Shh is a pleiotropic morphogen that can trigger many different cellular outcomes. During CNS development, multiple morphogen signaling domains can overlap and interplay to trigger cellular responses. For instance, Shh activity can be modulated by FGF, Wnt, or BMP but this crosstalk mainly relies on pathway transduction mechanisms rather than on modification of Shh gradient distribution [23,24,25,26,27,28,29,30]. In the context of the establishment of the Shh gradient itself, it is critical to understand how Shh protein is secreted and, more importantly, transported to induce many processes at long ranges. In this review, we tackle these questions: Where are the sources? How is the protein distributed? How are the distances managed? Are Shh gradients actually present in neuronal tissues? What are the consequences of disrupted Shh delivery?

First, we describe the different sources of Shh, which represent crucial signaling centers in the CNS at various stages of life. Then, we explore the different mechanisms and strategies by which Shh can diffuse and move away from these sources. Finally, we discuss how perturbation of Shh concentrations can be linked to several neural disorders.

## 2. Shh Sources in the Vertebrate Neural Axis

### 2.1. Neural Tube

Historically, the first Shh sources identified during vertebrate development are the floor plate, the notochord, and the zone of polarizing activity (ZPA) at the posterior side of the developing limb [31,32]. Here, we focus on the nervous system, and we will not address the ZPA but advise readers to explore other very detailed reviews on the subject [14].

In addition to the body of work focused on Shh diffusion during Drosophila wing disk development, the gradient of concentration induced by the floor plate and the notochord in vertebrate is the most studied. The text-book principle is that the Shh proteins secreted by the floor plate and notochord will diffuse throughout the ventral half of the developing neural tube, creating a gradient of concentration along the dorsoventral axis. Floor plate cells themselves will need high levels of Shh at a precise stage before inducing a gradient, which is required for them to gain their identity [33]. Shh expression by floor plate cells is also regulated by other important morphogens, such as BMPs. Indeed, transplantation of BMP-expressing cells next to the floor plate inhibits Shh expression [34], whereas inhibition of BMPs via the addition of Chordin significantly increases Shh expression at the floor plate [35]. Taken together, these data highlight how other morphogens are able to control the expression of Shh and ultimately affect the gradient distribution of the secreted protein. According to their location within this concentration gradient, neural progenitors will acquire precise cell identities. This idea was developed from in vitro experiments in which neural progenitors were exposed to different concentrations of Shh [5,36,37]. In the ventral spinal cord, five groups of neurons are identified thanks to a combination of transcription factors and their final target projections in the nervous system. In the V3 domain, neuron progenitors express *Nkx2.2* and will give birth to commissural neurons residing at the extreme ventral side of the neural tube, just next to the floor plate. Motoneuron progenitors express Olig2 and project towards the muscle in the periphery. V2, V1, and V0 progenitors (Figure 2A) each express a different set of transcription factors [7]. This patterning is dependent on the Shh signal, since its disruption in neural progenitors away from its source perturbs the neuronal pool patterning [13,38]. Both the concentration and the time of exposure are critical parameters to consider [39]. Moreover, the Shh pathway is unique in a way that Ptch1 induction by Shh leads to a double feedback loop in receiving cells, allowing a refinement of the gradient of signaling responses [40]. Overall, this precise patterning delimitation requires gene regulations organized in a network of transcription factors [41]. Morphogenic gradients are essential for neuronal identity and organization in the neural tube, but it is important to note that cell intrinsic features are also at play to regulate the sensitivity to these extracellular factors. For instance, by comparing several species, Uygur and collaborators identified cell intrinsic downstream mechanisms regulating Shh sensitivity [42]. By distinctively regulating the response to extracellular Shh, different species are able to adjust the production of the specific patterns according to their size. How the boundaries between different cell populations are defined during development according to a potentially noisy and variable gradient of diffusible proteins has raised many interests. For instance, the antiparallel gradient of Shh and BMPs (secreted from the roof plate) act together to induce more precision into the establishment of delimited neuronal territories [43]. The crosstalk between multiple morphogens is considered a key process for defining highly specific domains along the dorsoventral axis. However, a recent study based on mathematical modeling suggests that the variability of a single morphogen gradient is consistent with the production of a defined pattern in the neural tube. Therefore, a single morphogen, such as Shh, should be sufficient to induce the required precision for neural population delineations [44].

All of these studies mainly use the expression of target genes as read-outs for the activity of Shh; however, the direct visualization of Shh gradients has rarely been performed. A transgenic mouse was produced with green fluorescent protein (GFP) fused to Shh (Shh-GFP) without blocking its function [45]. This fusion protein shows a punctate fluorescent signal at the apical membrane of progenitors along the ventricle. This supports the idea that Shh is indeed secreted in the embryonic cerebrospinal fluid in the ventricles and may be bound at the cell surface of progenitors lying in the ventricle border; however, how Shh can signal throughout the width of the tissue is still not understood. Not much of a signal was detected inside the neural tube tissues using the *Shh-GFP* mice. When the Shh gradient was investigated by immunofluorescence, a gradient can be observed [46,47]. Most of the signal is concentrated at the notochord and floor plate. At embryonic day 10.5 (E10.5), the Shh distribution can be virtually divided into two sections in the rest of the neural tube. Within the first 50 μm away from the floor plate, the fractional change in the gradient is high, whereas it is very low for the rest of the dorsoventral axis of the neural tube [47]. Importantly, in the first images from Gritli-Linde and collaborators, we could see some Shh proteins transported away from the neural tube into the surrounding tissue and especially into the sclerotome [46]. Only recently this observation was confirmed, and the precise role for this release of Shh outside the neural tube was addressed [48]. Decreasing Shh quantity in the sclerotome in the mesoderm reduces the number of motoneuron progenitors and mature motoneurons, showing a role for extra neural tube Shh in neuronal patterning. 

Floor plate-derived Shh is also an attractive signal for commissural neurons in the developing spinal cord [49]. During development, neurons establish their long-range connections with specific targets at remote locations in the nervous system. This process is mediated by extracellular cues, i.e., secreted proteins which can guide growing axons. Shh attractive response relies on the Boc and Ptch1 receptors [15,50], Smo activity [51], the Dock/ELMO complex and Arl13b activity [52,53]. It is important to note that this guidance response is switched to a repulsive response to Shh once the axons have crossed the midline [54]. 

All of these studies focused on the downstream signaling pathways triggered by Shh. Here, we discuss how Shh is displayed as a signaling gradient in the tissue to induce such a guided response. The first studies showed that Shh can act as a chemoattractant on axonal growth cones [49,51]. In this context, Shh was either directly applied in the cell media as a gradient or by exposing neurons to secreting cells (transfected cell lines or notochord explants). Nonetheless, this model does not reproduce the Shh diffusion that occurs in a developing tissue. First, there are several steric constrains due to the tissue cell compaction. Second, Shh has a palmitoyl moiety added at its N-terminal cysteine and a cholesterol at the C-terminal extremity, which probably increases its interactions with cell membranes. When Shh distribution was analyzed in the neural tube by immunofluorescence, it appears that the gradient is biphasic [47]. In regions very close to the floor plate, Shh concentrations are very high, whereas the Shh gradient is mostly shallow throughout the dorsoventral axis of the neural tube. This contrasts with the in vitro models used to investigate axonal responses to Shh where gradients are linear. In vivo, commissural axon guidance is not only performed by Shh but relies on a synergy with netrin-1 [47], which compensates for this shallow gradient. Netrin-1 distribution in the neural tube has also been questioned. Midline-derived netrin-1 is dispensable for commissural axon guidance in the hindbrain [55], where the actual important source is ventricular zone progenitors. In the spinal cord, guidance defects are subtler revealing a difference between the two regions and highlighting the importance of ventricular-derived netrin [56]. Netrin1 is highly expressed by progenitors, and the protein is transported along the glial fibers to be presented along the path of growing axons at the pial surface. Ultimately, both sources are required for complete midline crossing in the neural tube [57]. The example of netrin1 illustrates the two different mechanisms allowing axon guidance: chemotaxis and haptotaxis. Chemotaxis, the first process identified, acted at long distances and was reproduced in in vitro cell culture assays. Secondly, haptotaxis is a guidance process relying on cell–cell close interactions and adhesions. In the light of these observations and the peculiar post-translational modifications of Shh, one can easily wonder whether Shh could induce haptotaxis responses too; however, the cellular sources of Shh are more limited in the neural tube than they are for netrin. Shh is only expressed by the floor plate and the notochord. Moreover, in vitro assays showed that netrin can induce outgrowth when bound to a substrate, whereas Shh cannot [58]. This suggests that Shh induces only chemotaxis and, therefore, acts at long range. Again, these observations pinpoint the need for Shh to have strategies to shield its lipophilic properties and diffuse. In that last study, only outgrowth was tested using a matrix-bound Shh. Previous data also indicated that Shh cannot induce axonal growth but only guidance using soluble proteins [49]. The way that axon guidance, i.e., directed growth, is impacted by Shh when it is bound to cell membranes or matrix remains to be determined.

### 2.2. Anterior Shh Sources in the Neural Axis 

In the anterior part of the developing neural axis, Shh is first expressed in the prechordal plate. Shh is required for the survival of the cells that form this important signaling center [59] and subsequently, also induces ventralization of the neural structures located just above the prechordal plate [60,61]. For example, ablation of this structure induces cyclopia, similar to Shh deficiency [62]. This induction is performed in coordination with BMP7 to induce rostral diencephalic ventral midline [63]. This is one crucial difference compared to what happens in the neural tube in which Shh and BMPs antagonizes each other. Shh expression is then triggered in the developing ventral neural tissue in the brain. Interestingly, different specific enhancers control Shh expression in the developing brain compared to the prechordal plate [60]. One enhancer has been identified as being required for prechordal plate expression and also for Shh induction in the ventral forebrain midline (namely: SBE7; [64]). Disruption of Shh expression by CRISPR/Cas9-mediated knockdown of this activator induces severe cranial and brain malformations typical of holoprosencephaly.

Shh is also expressed directly in the diencephalon, midbrain, and hindbrain (Figure 2A, [65]). The size of these structures as well as their anteroposterior and dorsoventral patterning are also regulated by Shh [66,67]. BrdU incorporation assays and cyclinD1 expression analysis in *Shh* mutants demonstrate the requirement of Shh on cell proliferation in the diencephalon and midbrain. Although the action of Shh is direct in the ventral part of the neural axis, it appears that it can also indirectly regulate proliferation in the dorsal parts of the diencephalon. This is possible through the regulation of other signaling pathways, including Wnts and FGFs [66]. The zona limitans intrathalamica (ZLI) is an important signaling center which lies in the diencephalon during development. Shh expression has been described in the ZLI in the neuroepithelium [68]. Using *in ovo* electroporation in the chick embryo, it has been shown that Shh from the ZLI regulates diencephalic regional identity in the forebrain [69]. Later on, this role was confirmed in the mouse and, more precisely, in the formation of the thalamus, via the induction of the *Gbx2* gene [65], and hypothalamus [67]. All of these expression data were obtained by *in situ* hybridization, but more recently, the Shh protein was also shown by immunofluorescence in the ZLI [70]. In mouse embryos mutant for *Rpgrip1l* (a causative gene in severe human ciliopathies) in which primary cilia are disrupted, Shh expression is abolished only in specific areas of the neural axis, including the diencephalon, but with persistent expression in the ZLI [71]. Interestingly, the mutants show a disruption of the diencephalic nuclei with an overall reduction in the ventral hypothalamus and axonal organization. This really highlights the importance of Shh production and thus signaling for the proper patterning of ventral structures at different levels of the developing brain, from the telencephalon to more caudal regions such as the midbrain.

Shh is involved in the patterning of the ventral telencephalon and directly regulates the expression of transcription factors required to specify ventral cell fates such as Nkx2.1 [72,73,74]. Results obtain from the analysis of *Shh-null* mice also support a role for Shh in regional specification of the ventral telencephalon [31,75]. Indeed, *Shh* mutants lack any sign of medial ganglionic eminence (MGE) and preoptic area (POA) development, two major sources of interneurons. Interestingly, blocking the cholesterol synthesis also induces a loss of ventral identity in the developing brain by interfering with Shh function [76]. Later during development, Shh is expressed by ventral structures themselves. At first (E10.5), Shh is expressed in progenitors within the ventricular zone of the MGE and POA [77]. There, Shh promotes the expression of *Nkx2.1* in proliferating cells [78]. Later on (E11.5), the pattern of expression in the MGE switches to the mantle zone, where interneurons are being differentiated (Figure 2A) [77,79,80]. One study suggested that the deletion of Shh in early postmitotic interneurons impairs the survival and proliferation of Nkx2.1 progenitors as well as the generation of oligodendrocytes [79,81]. Moreover, recent single RNA-seq data indicate that post-mitotic interneurons actually lose their Shh expression as they differentiate and reach the developing cortex [82]. However, interneurons in which ciliary proteins have been disrupted, such as Kif3A or Ift88 (i.e., defective in their responsiveness to Shh) show decreased migration velocity and fail to properly exit their tangential stream towards the cortical plate [83]. This suggests that Shh signaling is maintained in the cortex despite the absence of active *Shh* transcription. If and how Shh is diffused from MGE-derived interneurons is still unknown: post-mitotic neurons could still bear Shh at their membrane and transport it, even though they do not express it anymore. Since the cell bodies of the Shh positive cells seem to be far away from the ventricle surface, we cannot exclude that some processes could be extended to the surface. The use of fluorescent-tagged Shh or a clear immunofluorescent characterization of the Shh protein distribution is lacking to help us understand the function of Shh secretion within the developing ventral forebrain. 

With the expansion of the brain organoids field, several factors are now used to induce patterning in these models. Shh is one of the components required in several protocols to induce ventralization of the identity of human pluripotent stem cells [84]. Shh is required at early stages to induce a ventral phenotype (expression of *Nkx2.1*, for instance) in brain organoids. Moreover, exposure to high concentrations of Shh during neural maturation induces an increase in the number of somatostatin-positive neurons in brain organoids [85]. Shh can actually be replaced by directly activating the pathway via Smo, using the Smo agonist (SAG) to generate subpallium-like organoids from human cells [86]. The same strategy has been used previously to induce hypothalamic identity in neuronal organoids [87]. The protocols are now refined so that we can generate specific hypothalamic nuclei using a precise combination of factors, including recombinant Shh and Smo agonists [88]. The main caveat of these strategies is that the formation of cerebral organoids does not reproduce the precise regionalization of the brain. These protocols use Shh signaling to induce a global ventralization of the organoid, but they cannot provide multiple regions forming from one common cell assembly, such as what happens during development. To overcome this problem, a recent study developed a strategy in which Shh is not applied in the culture media but as a signaling center [89]. To do so, induced pluripotent stem cells (iPSC) expressing Shh were embedded in a larger organoid. With this approach, Shh should produce a gradient of concentration within the organoid. This experiment showed that classical Shh target genes, which are normally patterned in vivo by a gradient of Shh (such as *Nkx2.1* or *2.2*), are induced in the vicinity of the cellular organizer, whereas genes such as *Pax6* are repressed and remain expressed only in regions far from the Shh-organizer in the organoid. In the Shh-patterned area of the organoid, some hypothalamic-like cells can be detected, which express genes such as *LHX6*, *OTP*, *POMC*, and *TH*. Interneurons can also be produced this way. Thus, this is a clever approach to reproduce Shh-induced regionalization using human iPSCs. This strategy could be useful to analyze molecular and cellular mechanisms in the formation of morphogen gradients in a human-like context.

### 2.3. Embryonic Cerebrospinal Fluid

Histological analyses [90,91] and more recent transcriptomic data [92] have also identified the choroid plexus (ChP) as a major source of Shh during early development. The ChPs are secretory and highly vascularized tissues present in each one of the brain’s ventricles (Figure 2B). These structures are responsible for the production of CSF and secrete a wide panel of signaling factors during critical stages of neural development. Chronologically, the hindbrain ChP (hChP), present in the fourth ventricle, forms first, followed by the development of the two lateral ChPs and, finally, the one present in the third ventricle [93]. At the cellular level, the ChP consist primarily of a long sheet of multiciliated epithelial cells which form the blood-CSF barrier and maintain CSF homeostasis. However, other important cell types, such as mesenchymal and endothelial cells, are also present in this tissue. 

Interestingly, studies have shown that transcription and secretion by the ChP are both cell type-specific and spatially heterogeneous [94]. Thus, each ChP has its own specific molecular signature determined by its origin and final position along the body axis. These findings were supported by recent single cell and nuclei RNA sequencing performed on ChPs collected from each ventricle of the developing, adult, and aging mouse brain [92]. This single-cell temporal and spatial atlas revealed that *Shh* is expressed specifically during embryonic stages by the epithelial cells of the hChP. Moreover, isolation of embryonic CSF and ELISA experiments confirmed that Shh is secreted into the circulating CSF and can be detected as early as embryonic day E10.5 [95,96]. Consistent with its early expression pattern, Shh has been linked to hChP development, mainly by promoting vascularization and proliferation of ChP progenitor cells [91,97,98].

In addition, it is well established that Shh exerts important regulatory functions in cerebellar progenitor cells. Both Gli1 and Ptch1 expression are visible in the cerebellar ventricular zone (VZ) by E14.5, indicating active Shh signaling [96]. However, at this early stage, endogenous *Shh* transcription is undetectable in the developing tissue [99], suggesting a potential role of the transventricular Shh secreted by the nearby hChP (Figure 2B). Huang and colleagues were the first ones to show that hChP-derived Shh has a proliferative effect on radial glial cells and inhibitory interneuron precursors found in the cerebellar VZ [96,100]. As discussed in Section 2.4, Shh is later endogenously expressed by Purkinje cells [101,102]. The extent to which transventricular Shh affects other surrounding tissues is currently unclear. Several lines of evidence support a role for Shh in regulating progenitor cell proliferation and differentiation during cortical development [103,104]. Indeed, conditional deletion of Smo in nestin+ progenitors present in the neocortical VZ leads to a significant decrease in proliferation and an overall reduction in the size of the telencephalon at E18.5 [105,106]; however, in this case, the exact source of the Shh ligand still remains elusive as Shh mRNA is undetectable in the neocortex during this critical period of neurogenesis. Considering progenitor cells line the ventricular wall and their primary cilia protrude into the CSF, it would be interesting to investigate the potential implication of transventricular Shh derived from the hChP during corticogenesis. 

### 2.4. Shh Sources in the Adult Brain

In this review, we decided to focus on how Shh establishes gradients and diffuses away from its sources in the context of neural development; however, it is important to note that Shh also has important roles in the nervous system during adulthood. In this paragraph, we briefly give an overview of the different sources in the adult nervous system. 

The distribution of Shh protein- and mRNA-expressing cells have been carefully detailed in previous studies [107,108]. One important source of Shh is Purkinje cells, which produce Shh at early postnatal ages [102]. This endogenous Shh plays a key role in the proliferation and differentiation of granule cell progenitors, which will later give rise to all cerebellar excitatory neurons [109]. Later on, Shh signaling is one of the important regulators of adult neurogenesis in the hippocampus [106,110,111] and in the subventricular zone [112,113,114]. Moreover, Shh expression has been described in corticofugal neurons in the cortex [115] and is essential for synapse formation with local and callosal neurons. Recently neuron-derived Shh has been implicated in astrocyte morphogenesis and, in particular, in astrocyte ensheetment of neurons [116]. More is discussed on Shh function and traffic in neurons in the Section 3.4 below. Finally, oligodendrocytes are also a main source of Shh proteins within the adult brain [117]. The distribution of these Shh+ oligodendrocytes is widespread in the brain and could be an important source of Shh, potentially implicated in several functions, such as neuron, astrocyte, or oligodendrocyte homeostasis. The function of adult Shh is also highlighted by its implication in several diseases as described in Section 4 of this review.

## 3. Shh Transport

### 3.1. Extracellular Vesicles and Multimers

Here, we will explore the molecular mechanisms and strategies by which Shh can be diffused and moved away from its different sources. Theoretically, lipidated proteins, such as Shh, are lipophilic, making it harder for them to navigate the aqueous extracellular matrix. One way to overcome this limitation is to shield the lipid modifications of the protein from the surrounding hydrophilic environment, thus improving solubility and long-range diffusion. For instance, chaperon protein Scube2 can bind to the cholesterol modification (Figure 3A) and facilitate Shh secretion by increasing its stability in the extracellular space [118,119,120,121].

Alternatively, the shielding effect can be achieved by incorporating the membrane of supramolecular structures, such as lipoprotein particles or extracellular vesicles (EVs). Indeed, pioneer work revealed the existence of both lipoprotein and vesicular-associated forms of Hedgehog in Drosophila [122,123]. Lipoproteins are small particles surrounded by a single phospholipid outer shell (Figure 3A) and are mainly involved in the extracellular transport of cholesterol, lipids, and fatty acids [124]. Interestingly, in vitro experiments showed that vertebrate Shh can be found in purified lipoprotein fractions [125] and also be efficiently coimmunoprecipitated with apolipoproteins, which are major lipoprotein scaffold proteins; however, if and how Shh-positive lipoproteins regulate Shh signaling in vivo is currently unclear.

Moreover, strong evidence suggests that Shh can also be efficiently released from cells loaded in EVs [126,127,128]. Contrary to lipoprotein particles, EVs are surrounded by a phospholipid bilayer and are produced by Shh-expressing cells themselves (Figure 3A). Studies performed on both Drosophila and mammalian systems showed that the release of Shh in these vesicles is dependent on the endosomal sorting complex required for transport (ESCRT) machinery. This machinery regulates several membrane remodeling processes, such as the production and release of two different types of EVs: ectosomes, which are formed by direct cell surface budding, and exosomes, which are released from the cell by fusion of intracellular multivesicular bodies (MVBs) with the plasma membrane [129]. Interestingly, electron microscopy showed that Shh accumulates in MVBs, suggesting it is released in exosomes rather than ectosomes [127,128]. More importantly, these Shh-positive exosomes are able to regulate the Shh signaling pathway both in vitro [128] and in vivo [126]. In their study, Coulter and colleagues showed that ESCRT member Chmp1a (Charged Multivesicular Body Protein 1A) is required for the proper release of Shh-positive exosomes by the hChP and Purkinje cells during development. They observed that a *Chmp1a* loss-of-function reduces the formation of intraluminal vesicles within MVBs and disrupts secretion of Shh-bound vesicles. Consistently, *Chmp1a null* mice embryos have significantly lower Shh protein levels in their CSF compared to controls and display defects associated with impaired Shh activity, such as decreased proliferation of cortical and cerebellar progenitor cells. These findings provide the first evidence for a previously undescribed role of secreted extra-vesicular Shh in forebrain and hindbrain development.

Yet another way of shielding the lipid modifications from the aqueous environment involves the formation of multimers (Figure 3A). It has been shown that differently labeled Shh isoforms can be coimmunoprecipitated, suggesting that the protein is able to form homomultimers [130]. Interestingly, biochemical studies indicate that the addition of cholesterol and palmitic acid are both required for the generation of these multimeric forms [130,131,132]. Consistently, the lack of lipid modifications prevents multimerization and reduces the range of diffusion and the activity of Shh in vivo [131].

### 3.2. Membrane-Bound Regulators

Hedgehog interacting protein (Hhip) is a glycoprotein attached to the cell membrane that is able to bind to all of the mammalian hedgehog proteins with high affinity (Figure 3B; [133]). During nervous system development, *Hhip* mRNA is detected at early embryonic stages in areas surrounding Shh-expressing cells. Indeed, Hhip is expressed in the ventral midline and in the ventromedial somites surrounding the Shh-expressing notochord as well as in ventrolateral cells of the floor plate [133]. In the developing brain, Hhip expression mainly overlaps with Ptch1 and is detected in the neuroepithelium of the third ventricle and in the medial region at the MGE/LGE boundary [134]. Since Hhip is expressed downstream of Shh activation and can bind Shh itself, it was proposed to act as a negative feedback antagonist. In vitro studies first suggest that Hhip might antagonize Shh signaling by sequestrating it at the membrane, reducing its availability in the extracellular milieu [130,133]. 

During neural tube patterning, Hhip and Pct1 mediate ligand-dependent antagonism (LDA) to Shh. Indeed, Hhip deficiency increases Shh signaling and ventralization of the neural tube [135,136]. Unlike Ptch1, which promotes LDA through endocytosis of extracellular Shh, Hhip mediates LDA by sequestrating Shh at the cell surface to prevent ligand/receptor interaction (Figure 3B) [136]. Structural analysis of the Hhip structure revealed that the β-propeller domain present in the Hhip ectodomain is responsible for Shh binding [137,138]. Hhip does not only enable feedback for Shh antagonism at the cell surface of responding cells but can also provide non-cell autonomous inhibition when secreted in the extracellular space [139]. This distant inhibition was demonstrated in the neural tube, where ectopic dorsal expression of Hhip1 repressed both Nkx6.1 expression and mitosis in ventral cells that do not overexpress the transgene. Additionally, ectopic expression of a membrane-tethered Hhip1 did not affect neural tube patterning, supporting a mechanism by which secreted Hhip1 is involved in long-range Shh inhibition. Distant Shh pathway antagonism requires the cell-surface retention of secreted Hhip by heparan sulfates (HS). Thus, secreted Hhip accumulates at HS-containing neuroepithelial basement membranes to regulate Shh ligand distribution along the neural tube [139]. Interestingly, Hhip is also expressed in commissural axons to promote Shh-mediated chemorepulsion of post-crossing axons. Indeed, Hhip-induced expression downstream of Shh seems to regulate the switch between Shh-mediated chemoattraction and chemorepulsion on commissural axons [140]. Altogether, these studies show that both membrane-tethered and secreted Hhip regulates the Shh gradient and signaling at both short and long range by sequestrating the Shh ligand. Importantly, Hhip provides a feedback inhibition in response to the Shh ligand to enable a spatiotemporal regulation of Hedgehog responses.

Other extracellular matrix molecules and cell-surface proteoglycans can interact with morphogens to regulate gradient formation and signaling efficiency (Figure 3B). Among these molecules, heparan sulfate proteoglycan (HSPG) are highly expressed during embryonic development and have already been described as important regulators of morphogen activity, including Hedgehog [4,141,142,143]. Cell-surface HSPG carry HS glycosaminoglycan (GAG) chains (Figure 3B) that are covalently linked to a glycosylphosphatidylinositol-anchored or transmembrane core protein for glypicans and syndecans, respectively [144]. The first evidence of Hedgehog (Hh) gradient modulation by HSPG has been described in the Drosophila wing imaginal disk. In these studies, mutation of HS-polymerizing enzyme was shown to disrupt Hh diffusion and target gene expression in Hh-responding cells located at the antero-posterior border and the anterior compartment [145,146]. Although HS was shown to promote Hh activity in Drosophila, contradictory results have been described in mammalian cells. Indeed, in mosaic embryonic bodies comprised of both Shh-producing cells and Shh-receiving cells, Shh diffusion and signaling increases when receiving cells are HS-deficient [147]. Interestingly, this study also demonstrates that HSPG can inhibit Shh responses in a non-cell autonomous manner. The discrepancies in HS-dependent modulation of Hedgehog between Drosophila and mammals may rely on different transport mechanisms. In Drosophila, Hh transport can be mediated by direct cell–cell contact through cytonemes stabilized by HSPG [145,147,148]. Thus, HSPG might differentially affect Hh trafficking when located at discrete membrane domains involved in cell–cell interaction. Interestingly, several studies suggest that HSPG are more prone to influence Hedgehog responses in receiving cells since the removal of HS from producing cells does not affect Hh diffusion and activity [145,147,148]. 

Different glypican proteins, including Gpc1, Gpc3, Gpc5, and Gpc6, have been shown to interact with Shh to regulate its diffusion and signal transduction [147,149,150,151,152,153]. In vitro experiments conducted in cell lines revealed that Gpc can either enhance or inhibit Hedgehog signaling through different mechanisms. Indeed, Gpcs were proposed to regulate ligand binding at the cell surface, acting as co-receptors to modulate ligand/receptor interaction and endocytosis [150,151]. For instance, silencing *Gpc5* in rhabdomyosarcoma cell lines was shown to decrease the transcription of the Shh target gene *Gli1* [151] while depletion of *Gpc5* in embryoid bodies increases Shh long-range diffusion and the expression of its target genes *Olig2* and *Islet1* [147]. In the latter study, the authors showed that Gpc5 deficiency only in Shh-transporting cells is sufficient to increase Shh response in HS-competent responding cells. This result suggests that Gpc5 may inhibit Hedgehog long-range signaling in a non-cell autonomous manner, probably by sequestrating it; however, the specific contribution of Gpc5 expressed by Shh-responding cells has not been investigated [147]. Regarding Gpc function in Shh-responding cells, Li and collaborators showed that Gpc5 can bind Shh and facilitate the Shh-Ptch1 interaction to increase downstream pathway activation [151]. Therefore, Gpc could affect both gradient establishment and signal transduction when expressed in Shh-transporting cells or responding cells. Many studies have shown that the ability of Gpc to bind Shh and regulate its signaling rely on HS-GAG chains [149,151,152]. This mechanism is thought to depend on the interaction between the Cardin Weitraub (CW) motif of the Shh protein, which contains positively charged amino acids with negatively charged HS-GAG [154]. Nevertheless, other studies reported that GAGs are not required to mediate the Shh/Gpc interaction, suggesting that Shh can also directly bind Gpc core protein [150,153]. 

Interestingly, Gpc can also be released from the cell surface following cleavage of the glycosylphosphatidylinositol anchor by the lipase notum [155] or through proteolytic cleavage by furin-like convertase. Thus, Gpc shedding was proposed to regulate morphogen signaling either by decreasing morphogen binding at the cell surface or by sequestrating morphogens in the extracellular space [144,150]. For instance, Capurro and colleagues showed that proteolytic cleavage of Gpc3 is required for Shh inhibition. Conversely, convertase-resistant Gpc3 stimulate Shh signaling by increasing the Shh/Ptc1 interaction [156]. 

During cerebellum development, Gpc5 is expressed by granule precursors cells (GCP) and promote Shh binding and signaling. More precisely, HS-GAG carrying 2-O-sulfo-iduronic motifs are responsible for Shh binding and GCP proliferation [157]. 

In the neural tube, Gpc1 expression is detected in the floor plate, motor neurons, dorsal-root ganglionic neurons, and commissural axons and has been shown to regulate Shh-dependent axon guidance in a GAG-independent manner. Indeed, Gpc1 specifically expressed in commissural axons promote post-crossing commissural axon repulsion through the expression of Hhip downstream of Shh canonical pathway activation [153]. The sulfation pattern of HSPG is also likely to affect Shh-dependent neural tube patterning. Indeed, overexpression or knockdown of the endosulfatase Sulf1 affects both Shh diffusion and target genes expression [158,159].

Loss of function of glypicans or GAG-modifying enzymes leads to severe neurodevelopmental malformations, such as holoprosencephaly or microcephaly [160,161,162]. 

Therefore, the diffusion of the Shh gradient is not only regulated by the structure, quantity, or polymerization of the proteins but also by how they interact at the surface of cells. The panel of extracellular proteins able to regulate Shh diffusion establishes another level of regulation to refine the distribution of the signal. 

### 3.3. Cytonemes

As mentioned before, post-translational modifications of Shh by a cholesterol at its N-term domain and a palmitate at its C-term domain is likely to restrain its spreading in close proximity to Shh-producing cells [4,163]. In addition, these lipid moieties are likely to anchor Shh to membranes, supporting a trafficking mechanism mediated by direct cell–cell contact. Cell–cell-mediated Shh transport through specialized filopodia, called cytonemes, is one of the mechanisms proposed to explain long-range Shh signaling (Figure 3C). Cytonemes are dynamic actin-based filopodia with a diameter ranging from 20 to 200 nm, which can extend from the cell surface up to 700 microns [164,165]. These protrusions can be found either at the surface of morphogen-producing or -receiving cells to enable morphogen transport and signaling [166]. Cytonemes were initially described in the wing imaginal disk in Drosophila, where they emanate from wing disk cells and project in an oriented manner towards the antero–posterior border [165]. Evidence of cytoneme-dependent Hh transport was described both in Drosophila and vertebrates, where cytonemes can be carried either by Hh-expressing or Hh-responding cells [164,167,168,169]. Along cytonemes, the Hh protein does not travel through the membrane, but is transported intracellularly inside in extracellular vesicles originating from multivesicular bodies [122,164,168]. These exovesicles carrying anchored-Hh can also contain other Hh pathway components, such as Disp, Cdon, Drosophila ortholog Ihog, and the glypican Dlp, and are thought to mediate Hh release and signaling at cytoneme sites in a synapse-like manner [122,170]. 

Cytonemes responsible for Shh transport and signaling seem to display specific dynamics. In vitro experiments showed that cytonemes containing Shh or its co-receptors Cdo/Ihog and Boc are more stable, which could potentially facilitate Shh reception and signalling [122,167,168,169]. The transmembrane protein Dispatched (Disp), responsible for cholesterol-modified Hh release, also colocalizes with Hh at cytoneme sites (Figure 3C) and is required for Shh-induced cytoneme formation [164,171]. Interestingly, cytoneme formation can be modulated by Hh itself. Indeed, Shh overexpression in mammalian cells was shown to increase cytoneme occurrence [171]. Moreover, other Shh pathway components, such as Hh co-receptors Cdon/Ihog and Boc/Boi and the transmembrane protein Disp, also localized at cytonemes sites and were shown to regulate cytonemes. Indeed, overexpressing Cdon and Boc increase cytoneme occurrence and Shh-induced cytoneme formation [168].

Other extracellular matrix molecules, such as HSPG, are localized at cytoneme sites and can regulate both cytoneme dynamics and Hh gradients [148,164,172]. Indeed, expression of HS chains is necessary to stabilize Ihog-ovexpressing cytonemes in Drosophila [148]. In the wing disk, the HSPG glypicans Dally and Dally-like protein (Dlp) regulate cytonemes through the interaction with Ihog fibronectin domain (FN1) [148,167]. Overall, these studies suggest that both Hh pathway components and the extracellular matrix molecules are localized at the cytoneme interface to orchestrate Shh delivery at defined signaling domains.

In the central nervous system, cytoneme-like structures were only described in the neural tube, where they emanate from the Shh-expressing cells of the floor plate [173]. Recently, Hall and collaborators revealed that the genetic removal of the molecular actin motor myosin 10 disrupts Shh vesicular trafficking inside cytonemes and affects Shh signaling in the neural tube. Indeed, *Myo10* knockout mice showed reduced Shh domain in the floor plate associated with patterning defects which highlight the functional relevance of cytonemes in Shh gradient formation and activity [168]. Nevertheless, *Myo10* knockout mice did not exhibit phenotypes similar to *Shh*-deficient mice, such as holoprosencephaly, suggesting that other trafficking mechanisms or different molecular motors are involved in the vesicular trafficking of Shh in the brain [174].

Currently, no evidence of cytonemes has been described in the developing brain, potentially due to non-proper fixation methods or imaging resolution limits [171,173]. Nevertheless, both neural progenitors and post-mitotic neurons can exhibit filopodia structures with potential implications in signaling processes through direct cell–cell interaction [175,176]. For instance, both intermediate progenitors and radial glial cells can extend short-range filopodia tangentially to contact neighboring cells, while intermediate progenitors also display long-range filopodia projecting radially to contact radial glial cells [177]. Interestingly, these long-range filopodia carrying Notch ligand Dll1 were proposed to drive long-range Notch signaling between intermediate progenitors and apical progenitors [177]. Although filopodia have already been described to play key roles in the embryonic CNS, the presence of specialized filopodia, cytonemes, driving morphogen transport and signaling during brain development remains to be investigated.

### 3.4. Axonal Transport and Neuronal Communication

Neurons are not only sensitive to extracellular Shh but can also be a source of Shh themselves. As described in the first section of this review, several neuronal populations are identified as Shh-producing cells during embryonic development and during adulthood. Besides these expression patterns, how this neuron-derived Shh is addressed to the target cells remains unclear. Is the protein secreted from the soma or at the synapse? Can Shh be delivered all along the axonal shaft? Is Shh delivered at the synapse to act as a neurotransmitter?

Although not much has been investigated in regard to the roles of neuronal-derived Shh, some enlightening data have been gathered in the visual system. Traiffort et al. have shown that radioactive signal can be retrieved in the superior colliculus, after injection of a radio-labelled Shh in the eye [178]. This was the first evidence of a Shh axonal transport. In the developing retina, retinal ganglionic cells can project either to the contralateral or ipsilateral areas of the brain, namely the thalamus and the superior colliculus. Shh is actually expressed by contralateral-projecting retinal ganglionic cells and is required for several functions, such as proliferation and differentiation of these cells [179]. Moreover, Shh is required for the segregation of ipsilateral versus contralateral retinal projections in the brain [180]. This function has been recently linked to a trans-axonal repulsion between Shh-expressing contralateral axons and the responsive ipsilateral ones at the optic chiasm [181]. The transport of Shh that was observed previously using radioactivity was confirmed using retina *in utero* electroporation to express a fluorescent Phluorin-tagged Shh. Interestingly, the fluorescence was found all along the axonal tract and increased at the optic chiasm, where the repulsion occurs. This suggests a potential increased secretion at this precise location, although at this developmental stage axons are already extended further than the chiasm. Only few studies investigated the molecular pathways of Shh trafficking in neurons and have identified Sortilin as a regulator of Shh secretion [182,183]. These questions were discussed by Herrera and colleagues [184], who notably mentioned the potential influence of axonal defasciculation. The disassembly, or at least decompaction of axons that happens actively at the optic chiasm, could indeed help with Shh diffusion throughout this structure. In any case, this phenomenon is another example of a specific and, until now, unique way of diffusing Shh signal during neuronal development to actually trigger a very specific response. Very recently, retina-derived Shh was also shown to be responsible for astrocyte production of Fgf15, which finally will allow the migration of interneurons in the thalamus [185].

Another intriguing question is the role of the persistent expression of Shh in some populations of neurons during adulthood. As mentioned previously, Shh-expressing neurons can be found in several brain regions, such as the cerebellum, hippocampus, cortex, or more basal regions. We just discussed the possibility that Shh is secreted along axons, but another idea would be that Shh is released at the axon terminals. Several reports show evidence suggesting that Shh could be active at synapses. The application of Shh on brain slices modulate the activity of neurons from the ventrolateral nucleus tractus solitarius [186]. These neurons receive inputs from another nuclei, the dorsal vagal motor nucleus, which expresses Shh, suggesting a potential transport and synaptic release. Similarly, Shh application onto a dorsal vagal motor nucleus induces a decrease in neuronal activity [187]. These cells also receive inputs from Shh-expressing neurons in the globus pallidus [178], again suggesting a potential Shh transport and release; however, in this study, the direct neuronal Shh release from the dorsal vagal motor nucleus or the globus pallidus was not investigated. Similarly, in the cortex, a recent study showed that Shh signaling modulates GABAergic maturations [188]. In that study, the authors show that Shh protein is present in the somatosensory cortex by ELISA but without more precisely identifying the cellular source. All of the functional manipulations were focused and smoothened and, therefore, did not address the potential role of Shh directly but rather the pathway activation via Smo. One study knocked-out Shh in dopaminergic neurons using the Cre-lox system (*Dat-Cre*) [189]. In addition to a survival cell autonomous effect on dopaminergic neurons themselves, the authors showed that Shh triggers a response in the striatum, received by dopaminergic projections, which secretes Glial-cell-line-Derived Neurotrophic Factor as a survival factor for dopaminergic neurons. More recently, these results were debated, as another study did not reproduce the findings [190]. Using genetic fate mapping (using the *Shh-CreERT* mouse line), the authors show that perinatal or young adult TH-positive neurons are not generated from the Shh lineage. Moreover, deletion of the *Shh* gene in dopamine transporter-positive neurons does not affect the survival of nor induces a motor phenotype. These contradictory data show how the identification of Shh sources in the developing and adult brain is not really clear yet. In the last study, the authors suggest that Shh might come from the cerebrospinal fluid, as we discussed in Section 2.4. Finally, a report recently showed the presence of Shh at synapses in the CA1 stratum radiatum of the mouse hippocampus using electronic microscopy [191]. Shh present in EVs can also be detected in vitro in close proximity to hippocampal neurons [192]. Although some reports show that Shh can induce synapse swelling [193], further experiments are needed to understand if neuronal activity can trigger Shh release at the synapse and the function of this Shh-mediated neuronal communication. Further investigations are also needed to identify the precise molecular and cellular mechanisms underlying how Shh is transported along axons.

## 4. Shh Distribution Defects in Neurological Disorders

As described above, Shh ligands originating from different sources can be distributed through various mechanisms to trigger pleiotropic functions during CNS development. Abnormal production and delivery of Shh during development can impact various processes, such as proliferation, differentiation, neuronal migration, or axon guidance, that may lead to neurological disorders. In the adult nervous system, reactivation of the quiescent Shh activity can occur under pathological conditions to promote neuroprotection and recovery or contribute to uncontrolled responses as observed in tumors.

### 4.1. Neurodevelopmental Disorders

During development, Shh deficiency leads to severe brain malformations, such as holoprosencephaly (HPE), a midline defect due to an incomplete prosencephalic cleavage into right and left hemispheres [75,194,195]. Recently, synonymous single nucleotide variant analysis revealed variants which are enriched in holoprosencephalic patients [196]. These variants are associated with a decrease in the secretion of Shh in in vitro models. No defects in mRNA folding or splicing were detected; however, this study highlights the importance of codon usage in the onset of holoprosencephaly. Proteasome inhibition rescues this effect, showing how these variants impact the formation of unstable Shh proteins. The *Shh* gene is not the only gene affected in holoprosencephaly; several other ones related to its secretion and/or diffusion can also be mutated. Recently, a study investigated the oligogenic inheritance in holoprosencephaly, i.e., the inheritance mediated by a combination of mutations in only a few genes [197]. The study shows the relevance of genes such as *SCUBE2* or *NDST1* (N-Deacetylase and N-Sulfotransferase 1). Interestingly, mutant mice lacking Ndst1, an enzyme responsible for HSPG sulfation, exhibit malformations that resemble lobar and semi-lobar HPE, such as facial hypoplasia, reduced brain size, and median cleft [160]. In these mutants, the ability of Shh to bind GAG is reduced and associated with a reduced Ptch1 expression. In light of the established roles of proteoglycans in the regulation of Shh diffusion and signaling, this study suggests that impaired HSPG biosynthesis may potentially disrupt Shh functions during early development. 

In accordance with this, accumulation of GAG is observed in some lysosomal storage diseases (LSD), such as mucopolysaccharidosis type II (MPSII), and correlated with impaired Shh activity [198,199]. In a mouse model of Niemann-Pick type C (NPC) disease, an LSD caused by a mutation of *NPC1* or *NPC2* genes leading to intracellular accumulation of cholesterol and glycosphingolipids, cerebellum morphogenesis is impaired. In addition, Shh activity is decreased in the cerebellum, and primary cilium is altered in *Npc1*-deficient mice and patients’ fibroblasts [200]. In this model, pharmacological restoration of cholesterol homeostasis rescues both Shh signaling and ciliogenesis. Cholesterol was reported to participate in ciliogenesis [201] and hedgehog signaling efficiency depends on the cholesterol accessibility at the ciliary membrane [202]. In addition, cholesterol modifications were proposed to be required for multimer formation and Shh long-range signaling [130,131,203,204]. Hence, disturbed cholesterol metabolism in LSD could either directly affect Shh distribution or indirectly affect its reception at the primary cilium in the cerebellum.

Dysfunction of primary cilia due to ciliary gene mutations can lead to neurodevelopmental disorders, such as neural tube patterning defects or forebrain and cerebellar malformations [70]. Conditional knockout or mutations of genes involved in ciliary formation lead to forebrain defects [205,206], and affect Hedgehog functions [83,207]. Given that primary cilia from apical progenitors are facing the cerebrospinal fluid in the dorsal telencephalon, ciliopathies could also potentially impair signal transduction of CSF-derived Shh at the ventricular surface.

### 4.2. Cancer

Medulloblastoma is a type of malignant childhood brain tumor of the cerebellum due to abnormal proliferation of progenitors, which is linked to a dysregulation of Shh signaling in 28% of cases [208]. The Shh-medulloblastoma (SHH-MB) subgroups are linked to mutations or copy-number alterations in genes related to the Shh pathway, such as loss-of-function or deletion of Ptch1, leading to constitutive activation of the pathway. Although Shh pathway overactivation is clearly linked to medulloblastoma, the etiopathology of SHH-MB is mainly related to the mutation of Hedgehog pathway components rather than the amplification of the *SHH* gene, which represents only 3% of SHH-MB [208]. Nevertheless, upregulation of Shh expression was reported in various brain tumors, including glioblastomas [209,210]. In multiple gliomas, the Shh-Gli pathway was shown to stimulate cell proliferation to promote tumor growth [210]. Interestingly, the severity and therapeutic resistance of the various brain tumors is thought to be sustained by stem cell-like cells [211]. In glioblastoma, Shh is highly expressed in CD133+ glioma stem cell-like cells, and the depletion of Shh from cancerous cells was shown to reduce intracranial tumor growth in a xenograft model [212]. Similarly, blocking Shh pathway activation through Smo-antagonist inhibits the self-renewing ability of glioma stem cell-like cells and decreases tumor growth [209,212]. Hence, targeted inhibition of the Shh pathway in distinct cancer stem cell populations could represent a promising therapeutic strategy for glioblastoma treatment.

### 4.3. Brain Injury and Inflammation

Expression levels of Shh protein in the central nervous system are elevated during development and decrease in the adult parenchyma. However, Shh expression can be upregulated in adults under pathological conditions, such as brain injury or inflammation. For instance, Shh is upregulated in reactive astrocytes after hippocampal lesion, traumatic brain injury, stroke, or in inflammatory models [213,214,215,216,217,218,219]. In these models, it has been proposed that Shh pathway activation exerts a neuroprotective effect by maintaining the integrity of the blood–brain barrier (BBB) [213,215,219]. Indeed, the selective inactivation of the Shh pathway in astrocytes can decrease BBB permeability through downregulation of junctional protein expression in endothelial cells [213]. In line with its function regarding the BBB, Hh pathway activation reduces neuroinflammation and improves the neurological outcome [213,215,219]. It was also shown that intrathecal administration of Shh can improve the recovery of mice in a model of cerebral ischemia [220]. Upregulation of the Shh pathway also promote the proliferation of astrocytes, microglia, and oligodendrocyte precursors [214,217]. Following acute brain injury, reactive astrocytes not only increase their proliferation but can also exhibit stem cell-like behaviors in response to Shh [221]. These two cellular responses could either promote glial scar formation or increase neurogenesis, respectively, with opposite outcomes in regard to brain recovery.

In multiple sclerosis, Shh immunoreactivity is detected in remyelinating lesions in astrocytes, endothelial cells, and macrophages [218,222]. In a preclinical model of focal demyelination induced by lysolecithin injection in the corpus collosum (LPC), Ferent and colleagues reported that Shh expression is triggered in the oligodendrocyte lineage and promoted Hh pathway activation in both oligodendrocytes and microglia [113]. Following the cuprizone model, Gli1 is activated in oligodendrocyte precursor cells (OPC) which will be engaged in remyelination processes, suggesting that the Shh pathway may promote recovery [223]. Consistent with this, genetic overactivation of the Shh pathway in *Ptch1 +/−* mice reduces the clinical score of mice subjected to experimental autoimmune encephalopathy (EAE) [224]. Recently, Laouarem and colleagues showed that Smo activation in microglia is required to promote OPC differentiation during remyelination, suggesting a crosstalk between oligodendrocytes and microglia that support Shh functions [225]. Overexpression of the Shh ligand in the lesioned brain was also shown to increase OPC proliferation and decrease inflammation and astrogliosis, suggesting a neuroprotective function [113,224]. Interestingly, Macchi and colleagues demonstrated that HS immunoreactivity increases at the lesion site in the LPC model and suggested that HS concentrates Shh at the lesion site to promote its signaling activity [226]. Indeed, HS are necessary to promote Ptch1 expression in the brain parenchyma surrounding the lesion site. Consistent with this, ex vivo experiments revealed that only Shh carrying the CW sequence is able to bind to the parenchyma surrounding the lesion.

## 5. Challenges and Perspectives

During nervous system development, Shh functions rely on the production of the Shh ligand by different sources and its delivery to receiving cells via multiple transport mechanisms. Accumulating evidence has shown that Shh gradient establishment is far more complex than a simple diffusion from a ligand release in the extracellular space. Indeed, the theory of passive diffusion has recently been reconsidered using in vitro models [40]. In this review, we have described the different sources of Shh during neurodevelopment and the different known ways the protein can be distributed. Further investigations will be necessary to determine whether these distinct Shh delivery and trafficking ways are engaged from distinct cell sources and how they affect gradient formations. Direct Shh transport through cell–cell interaction (cytonemes or axonal vesicular transport) versus indirect transport (i.e., passive diffusion or EV transport) might differentially contribute to various Shh-dependent responses.

First, it is conceivable that these different trafficking mechanisms do not allow the delivery of the same amount of ligand. For instance, direct transport through cytonemes was only described between producing cells and adjacent receiving cells and could allow the delivery of a large amount of ligand in a very defined and targeted space. Conversely, extracellular vesicles carrying Shh might travel long distances in the CSF to reach target cells in remote regions which may receive less ligand. The relationship between different modes of diffusion and the downstream signaling pathway also remains to be explored. Proximal sources or transport through direct cell–cell interaction may affect canonical or non-canonical responses involved in distinct processes differently than CSF-derived Shh.

It is noteworthy that most of the studies investigating Shh transport mechanisms used canonical-related readouts of Shh pathway activation, which are mainly involved in proliferation, differentiation, and cell fate decisions. How neuronal migration or axon guidance is controlled by a specific Shh diffusion model is still unknown. Many questions are still to be answered: How long-range and how stable is the Shh distribution in a living and developing embryo? How are EVs delivered throughout the tissues? Is one way of distributing Shh associated with one specific source? Are producing cells able to use several distribution modalities?

Although the development of new experimental paradigms, such as organoids, will help us address some of these questions, a few technical challenges may be hard to overcome. For instance, disrupting a specific transport mechanism responsible for Shh delivery is difficult since it may also affect the transport of other proteins, such as morphogens and growth factors. In addition, it is possible that long-range delivery of Shh may rely on a combination of transport mechanisms. Nevertheless, a genetic approach could be used to determine how specific source cells can affect Shh-responses at short versus long distance in vivo.

Exploring how Shh gradients are formed and maintained via different modalities and from various sources is an exciting challenge. Importantly, answering these questions will not only improve our fundamental understanding of the rules of neurodevelopment but may also open the door to new therapeutic strategies.

## Figures and Tables

**Figure 1 cells-12-00225-f001:**
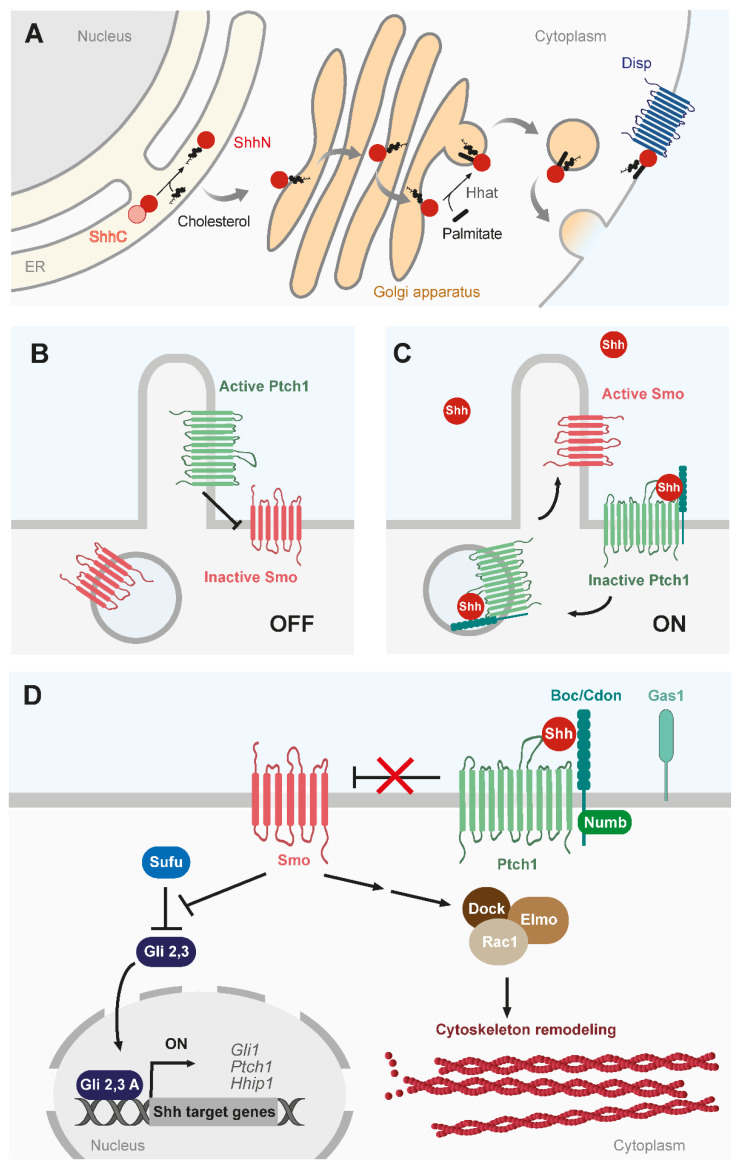
Shh signaling pathway. (**A**) Shh precursor is cleaved in the endoplasmic reticulum (ER). While the C-terminal part is discarded, the N-terminal part will be modified by the addition of a cholesterol molecule. Later, a palmitate is added to the peptide via the Hedgehog acyltransferase (Hhat). Finally, Shh is addressed to the plasma membrane where it interacts with Dispatched to initiate its secretion. (**B**) In the absence of Shh ligand, Ptch1 is localized at the primary cilium and inhibits the activity of Smo. (**C**) Upon Shh binding to its receptor Ptch1, Smo is activated and translocate to the primary cilium while ligand/receptors complexes are internalized. (**D**) Canonical pathway activation represses the sequestration of Gli2/3 in the cytoplasm by Suppressor of Fused (SUFU), allowing the transcription of Shh target genes. Shh pathway activation can also promote cytoskeleton remodeling downstream of Smo through the Elmo/Dock complex and Rac1 phosphorylation. The endocytic adaptor Numb is required for Boc endocytosis and Shh-mediated axon attraction. See main text for details. Shh, Sonic hedgehog; Ptch1, Patched 1; Smo, Smoothened; Cdon, Cell adhesion associated oncogene regulated protein; Boc, Brother of Cdon; Gas1, Growth-arrest specific gene 1; Suppressor of Fused, SUFU; Glioma-associated oncogene, Gli; Hedgehog interacting protein 1, Hhip1; Engulfment and Cell Motility, ELMO; Rac Family Small GTPase 1, Rac1; Dedicator of cytokinesis protein, Dock.

**Figure 2 cells-12-00225-f002:**
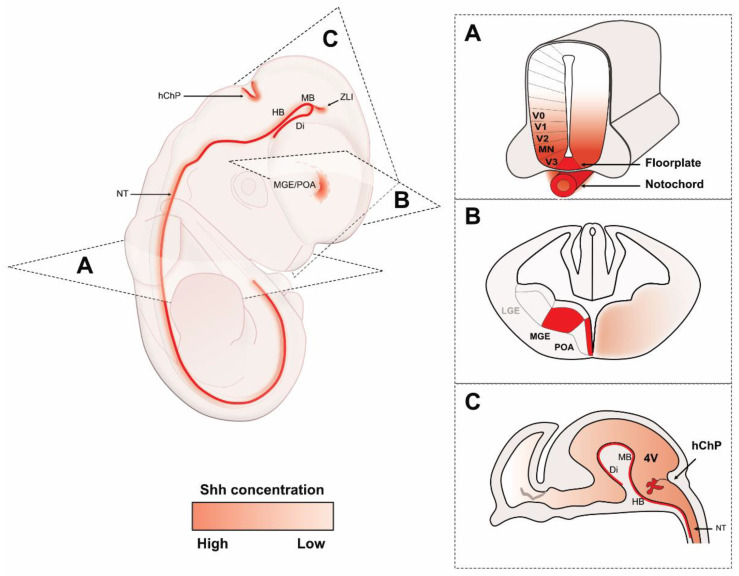
Schematic representation of Shh sources and gradients during central nervous system development. (**A**) Transversal section of the neural tube formation showing Shh expression (red) by the floorplate and the notochord. The ventral to dorsal Shh gradient defines the patterning of the ventral neural tube into 5 domains (V0, MN, V1, V2, V3). (**B**) During brain development, Shh is expressed by the MGE and POA in the ventral telencephalon. (**C**) Shh is also synthetized by the hindbrain choroid plexus (hChP) of the 4th ventricle (4V) and secreted in the CSF. Red-colored gradients represent the expected Shh diffusion from the different sources. NT: neural tube; MB: midbrain; HB: hindbrain; ZLI: zona limitans intrathalamica; Di: diencephalon; MGE: medial ganglionic eminence; LGE: lateral ganglionic eminence; POA: preoptic area.

**Figure 3 cells-12-00225-f003:**
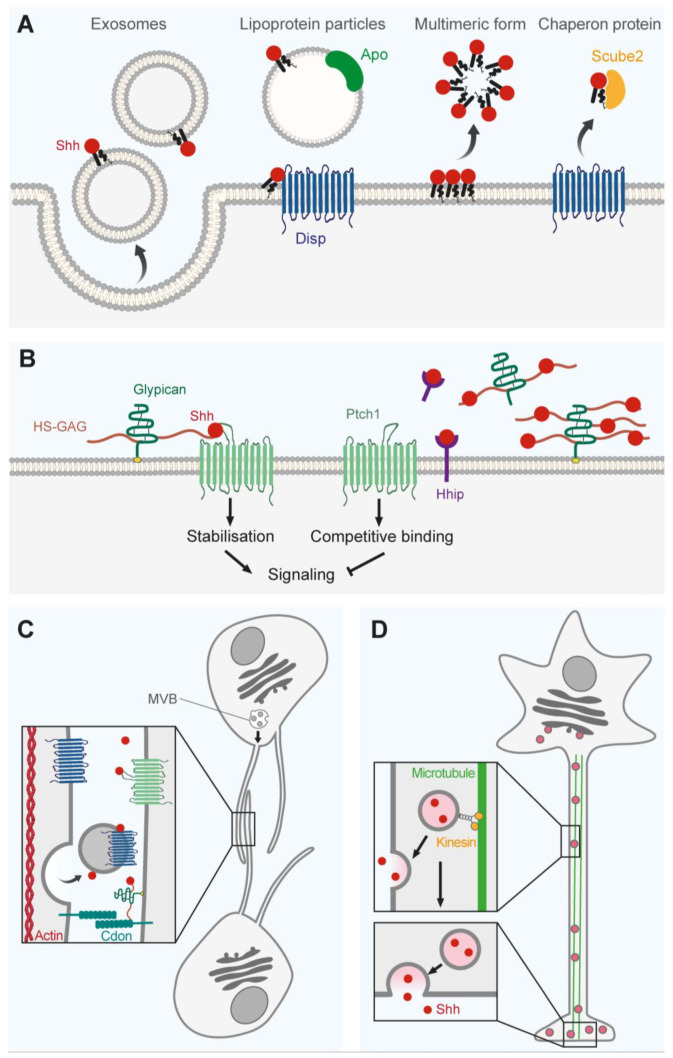
Transport mechanisms and cell-surface regulation of Hedgehog signaling. (**A**) Shh can be secreted out of cells by several mechanisms such as exosomes. Shh can also be transported by lipoprotein particles. Finally, the Shh peptide itself can be secreted directly by Dispatched at the membrane and subsequently form multimeric complexes or complexes with the chaperone protein Scube2. (**B**) At the cell surface glypicans can bind Shh through their GAG chains to stabilize ligand-receptor interaction and promote signaling. Conversely, both membrane-anchored and cleaved glypican and Hhip can sequestrate Shh preventing Hedgehog pathway activation. (**C**) Shh can also be delivered through direct cell–cell interaction via cytonemes. Shh is transported along cytonemes by exosomes originating from MVB. Dispatched and Ptch1 are localized at cytoneme sites to enable Shh release and signaling, respectively. Both Cdon and glypicans are required for cytoneme stabilization and Shh reception. (**D**) In neurons, Shh is transported along the axon and can be released at several sites: Shh can be secreted down the axon or directly at the synapse.

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
