# Peer review of "Establishing Hedgehog Gradients during Neural Development"

_cells, 2023, doi:10.3390/cells12020225_

Round 1

Reviewer 1 Report

In this manuscript, Douceau and collaborators provide a comprehensive review of the literature on establishing gradients of the Sonic Hedgehog (Shh) morphogen during neural development, with a focus on vertebrates. After introducing the notion of morphogen and briefly describing how Shh is produced, matured, secreted and internalized, three main chapters are covered: the Shh sources in the vertebrate neuraxis, the different mechanisms of Shh transport, and how Shh distribution defects can lead to neurological disorders. A last part presents the challenges and perspectives in the study of Shh transport. Overall, this review is very interesting and well documented and should be of great interest for the readers of the special issue of Cells entitled “Hedgehog signaling: advances in development and cancer”. The illustrations are clear and helpful. The English could certainly be improved and in many cases this would help to understand the author's arguments.

Here are a few more specific remarks. 

In Chapter 2.2 “Medial ganglionic eminence, preoptic area and zona limitans intrathalamica”:

-       The description of Shh spatiotemporal expression and function in the developing forebrain lacks precision. Shh expression in the prechordal plate, diencephalic floor plate and ventral hypothalamus are not even mentioned, although they play essential roles in patterning of the forebrain. Likewise, Shh expression domains are incomplete in Fig 2.

-       In this chapter the authors state that “Within the MGE Shh ligand is expressed in neurons” 242, while later in the paragraph they refer twice to Shh-expressing MGE progenitors (lines 252 and 255-256). This should be clarified.

Several sentences are unclear or too vague. A few examples :

-       P5 lines 161-167 “Although, a recent study…”: the idea here is very difficult to understand, even after looking at the cited paper.

-       P6 line 231 “Overall, this suggests that growth and guidance are two different processes.”… I don’t think this is what the authors meant…

-       P6 line 248, the term “ciliary defective interneurons” is not clear. Please be more precise.

Author Response

We would like to acknowledge reviewer1 for his/her constructive observations to our manuscript. We believe that the corrected version has improved its quality.

More precisely, here are our answers to each of the point raised by the reviewer1:

-First, we do agree that the section on the forebrain sources was not complete. Therefore, we added a paragraph describing the role o the prechordal plate in the induction of Shh sources in the forebrain at the beginning of section 2.2 (Lines 252-265 in the tracked version). Moreover, we also developed more the sections about the production and role of Shh in the diencephalon (Lines 294-301 and 309-316 in the tracked version). Overall these changes gave us the opportunity to cite more references important in this field, which we have missed in the first version. We have completed the Figure 2 accordingly. We also have changed the name of the section to “Anterior Shh sources in the neural axis” to include all parts.

-As far as we know, Shh expression within the MGE has not been clearly defined at the cellular subtype level and therefore its description lacks sometimes precision. From the in-situ hybridization data we have found in the literature, it seems that at early stages (E10.5), Shh can be expressed at the ventricular zone of the MGE, i.e. in the progenitor domain. Later on, during development (E11.5), the pattern of expression shows a strong signal at the center core of the MGE, where neurons are being differentiated and get ready to migrate out of the MGE (Flandin et al, 2011). However, recent RNA-seq data seem to show that post-mitotic neurons do not express Shh or only very low levels. We have made changes in the text to be clearer. You can see the text at lines 325-344 in the tracked version, corresponding to the part dedicated to the MGE, which we have moved after the part about the diencephalon.

-Previously P5 lines 161-167 / Now lines 171-176 in the tracked version:

We agree with the reviewer that the idea was not clearly explained in this paragraph. We have now modified the text to better describe the conclusion of the cited paper. The paragraph now ends like this:

“The interaction between multiple morphogens such as Shh and BMP in opposing gradients has been considered a key process for defining highly specific domains along the dorsoventral axis. However, a recent study based on mathematical modeling using available data suggests that the variability of a single morphogen gradient is consistent with the production of a defined pattern in the neural tube. Therefore, a single morphogen, such as Shh, should be sufficient to induce the required precision for neural population delineations.”

-Previously P6 line 231 / Now line 246 in the tracked version:

In this section, we wanted to describe the fact that Shh can only mediate chemotaxis and not promote axonal out growth per se. Since this difference is clearly mentioned and explained in the previous sentences of the paragraph, we believe that the sentence cited by the reviewer is not necessary, too general and adding confusion. We decided to remove it from our text.

-Previously P6 line 248/ Now lines 341-344 in the tracked version:

We agree that the ciliary defect is not specific enough, so we have added details of the genetic targets that were used in this study. This sentence has been modified as follows: “However, interneurons in which ciliary proteins have been disrupted, such as Kif3A or Ift88 (i.e., defective in their responsiveness to Shh) show decreased migration velocity and fail to properly exit their tangential stream towards the cortical plate”.

Reviewer 2 Report

Douceau S at al present a review of the Establishing Hedgehog gradients during neural development. The authors capitalize on the recent abundance of literature regarding the gradients of hedgehog signals, sources, and their role in normal development and disease.

I have some comments that the authors should take into account to avoid confusions.

Major

How the Shh gradient establishes in the presence of existing signaling molecules such as BMP, Wnt, and FGF is missing. What cross-talk partners are during neural tube formation or AP patterning? In my opinion, authors should add a section (possibly before section 4).

Minor

Introduction;

It should be concise; for example, lines 28 to 28 are unnecessary.

Lines 62 to 72 can be removed from section 1 (introduction) and added to section 3 (Shh transport).

Additionally, the title of section 3 (Shh transport) can be modified to ‘Shh processing and transport.’

Author Response

We would like to thank reviewer 2 for his ideas. We have taken his suggestions into consideration and here are our detailed responses:

Major comment:

We agree with reviewer 2 that interactions of the Shh gradient with other morphogens such as BMP, Wnt or FGF is a fundamental process during neural axis development. However, in this review, we decided to focus and explore in detail all the features and mechanisms related to the extracellular distribution of Shh protein in the extracellular space, from its sources, its transport and finally to its perturbations during pathology. As this is already a large topic and we really wanted to take the opportunity to develop as much as possible on this subject, we decided not to focus on the intracellular processes downstream of Shh activation.

Now, the effects of other morphogens interfere with Shh signaling mainly via regulation of the downstream signaling pathway with an end result in the regulation of genetic programs regulating cell fate, for example. Since this is an important topic to consider, as mentioned by reviewer 2, we decided to add a few sentences in the introduction to add this concept for the reader (lines 94-99 in the tracked version). At this point, we now cite some reviews covering the topic.

However, as we were reviewing this commentary, we realized that we were still missing studies in which the role of BMPs is explored in regulating the expression of Shh itself in the floor plate (Arkell and Beddington, 1997; Patten and Placzek, 2002). In this case, we believe that the results of this work fall within the scope of our review. One show that inhibition of BMP (via chordin) leads to an increase in Shh expression at the floor plate. This suggests that BMP may act as an inhibitor of Shh expression and, therefore, regulate the potential amount of protein produced by the floor plate. The other show that an ectopic source of BMP can reduce Shh expression in the floorplate. Thus, the ventral gradient of Shh in the neural tube must be modulated by BMPs.

We thank reviewer 2 for pointing us in this direction and have added these references and a few sentences in section 2.1 “Neural tube” (lines 138-144 in the tracked version).

Minor comments:

We have deleted line 28.

To be more concise and avoid redundancy with other sections of the review, we have deleted lines 70-74 in the tracked version. This information is also detailed in section 3.1.

We agree with reviewer 2 that the paragraph describing the role of Scube2 in Shh processing should be moved to the section dedicated to transport mechanisms. Therefore, we have moved the corresponding text to this section (now lines 452-454 in the tracked version). However, because we wish to focus on the extracellular mechanisms regulating Shh transport in the tissue environment, we decided not to address intracellular Shh processing in the same section and keep it apart in the general introduction. Moreover, moving this section would impact the balance in between figures (Fig1 and 3).

Round 2

Reviewer 2 Report

The revised version is acceptable